# Carborane-Containing Hydroxamate MMP Ligands for the Treatment of Tumors Using Boron Neutron Capture Therapy (BNCT): Efficacy without Tumor Cell Entry

**DOI:** 10.3390/ijms24086973

**Published:** 2023-04-09

**Authors:** Sebastian Flieger, Mao Takagaki, Natsuko Kondo, Marlon R. Lutz, Yash Gupta, Hiroki Ueda, Yoshinori Sakurai, Graham Moran, Prakasha Kempaiah, Narayan Hosmane, Minoru Suzuki, Daniel P. Becker

**Affiliations:** 1Department of Chemistry and Biochemistry, Loyola University Chicago, Chicago, IL 60660, USA; sflieger@luc.edu (S.F.); gmoran3@luc.edu (G.M.); 2Research Center for Nuclear Physics, Osaka University, 10-1 Mihoga-oka, Ibaraki-City 567-0047, Osaka, Japan; takagam@rcnp.osaka-u.ac.jp; 3Particle Radiation Oncology Research Center, Institute for Integrated Radiation and Nuclear Science, Kyoto University, 2-1010 Asashiro-Nishi, Kumatori, Sennan-gun 590-0494, Osaka, Japan; nkondo@rri.kyoto-u.ac.jp (N.K.);; 4Department of Medicine, Infectious Diseases, Mayo Clinic, Jacksonville, FL 32224, USA; yashodharmangupta@gmail.com (Y.G.); kempaiah.prakasha@mayo.edu (P.K.); 5Department of Chemistry and Biochemistry, Northern Illinois University, DeKalb, IL 60115, USA; hosmane@niu.edu

**Keywords:** boron neutron capture therapy, BNCT, matrix metalloproteinase, MMP, carborane, cancer, antitumor, boron delivery agents, binary radiation therapy

## Abstract

New carborane-bearing hydroxamate matrix metalloproteinase (MMP) ligands have been synthesized for boron neutron capture therapy (BNCT) with nanomolar potency against MMP-2, -9 and -13. New analogs are based on MMP inhibitor CGS-23023A, and two previously reported MMP ligands **1** (**B1**) and **2** (**B2**) were studied in vitro for BNCT activity. The boronated MMP ligands **1** and **2** showed high in vitro tumoricidal effects in an in vitro BNCT assay, exhibiting IC_50_ values for **1** and **2** of 2.04 × 10^−2^ mg/mL and 2.67 × 10^−2^ mg/mL, respectively. The relative killing effect of **1** to L-boronophenylalanine (BPA) is 0.82/0.27 = 3.0, and that of **2** is 0.82/0.32 = 2.6, whereas the relative killing effect of **4** is comparable to boronophenylalanine (BPA). The survival fraction of **1** and **2** in a pre-incubation boron concentration at 0.143 ppm ^10^B and 0.101 ppm ^10^B, respectively, were similar, and these results suggest that **1** and **2** are actively accumulated through attachment to the Squamous cell carcinoma (SCC)VII cells. Compounds **1** and **2** very effectively killed glioma U87 delta EGFR cells after BNCT. This study is noteworthy in demonstrating BNCT efficacy through binding to MMP enzymes overexpressed at the surface of the tumor cell without tumor cell penetration.

## 1. Introduction

Boron neutron capture therapy (BNCT) is a binary radiotherapeutic modality for cancer treatment, in which delivery agents containing ^10^B atoms are ideally concentrated in tumor cells and then irradiated with low-energy [1] (0.025 eV) thermal neutrons or epithermal neutrons (10,000 eV), which become thermalized as they penetrate tissues [2]. Once inside the tumor cell, the ^10^B nucleus adsorbs a neutron to form an excited state ^11^B isotope, which undergoes decay via nuclear fission [3]. As a result, the decay yields a high linear energy transfer (LET) α-particle (^4^He^2+^), as well as a ^7^Li^3+^ ion, each with high kinetic energy [4]. The highly charged particles have a kinetic range of approximately one cell diameter (5–9 μm), which limits the radiation damage to the cancer cell in which they arise, thus minimizing cytotoxic effects to the surrounding healthy tissue [5]. BNCT has shown great promise in clinical trials for treatment of glioblastoma multiforme, malignant gliomas, melanoma, as well as head, neck and lung cancer [6]. Currently, there are only two clinically investigated BNCT drugs, L-boronophenylalanine (L-BPA) and sodium borocaptate (BSH), which are neither tumor-specific, nor do they accumulate in high concentrations within the tumor cells. The effectiveness of BNCT therapy is governed by the selectivity of the drug and the specific accumulation of ^10^B atoms on tumor cells. In order for BNCT to be successful, it is necessary to deliver at least 10–30 μg of ^10^B atoms per gram of tumor (~10^9^ atoms/cell) and a sufficient amount of neutrons must penetrate and be absorbed by ^10^B atoms on the tumor cells to initiate fission of the resulting metastable ^11^B nuclei, which causes damage to the cell containing the irradiated boron atoms [7,8]. The advent of accelerator-based neutron sources (ABNS) has been encouraging for the overall promise of BNCT and breathes new life in BNCT research [9].

Carboranes are icosahedral boron cage molecules that are distinguished with a rich and diverse synthetic chemistry [5] and have shown promise in drug candidates for the treatment of various diseases. BSH is an icosahedral boron cluster molecule that has found use in treating various cancers through BCNT. BSH and L-BPA have demonstrated therapeutic efficacy in patients with high grade gliomas, recurrent tumors of the head and neck region, and also in patients with cutaneous and extracutaneous melanomas via BNCT [10].

Matrix metalloproteinases (MMPs) are a family of zinc-dependent endopeptidases that are involved in the remodeling and degradation of all components of the extracellular matrix (ECM) [11]. Under normal physiological conditions, MMP enzyme activities are well-regulated and play key roles in normal development, morphogenesis, bone remodeling, wound healing and angiogenesis [12,13]. However, abnormal levels of expressed MMP enzyme activity is implicated in a number of disease states, including degradation of articular cartilage in arthritis, tissue remodeling and weakening of left ventricular wall in congestive heart failure, and in tumor growth and metastasis including MMP-2, -9, and -13 [14,15,16,17,18,19], in addition to promoting tumor progression through degradation of the ECM. As a strategy to halt the progression of diseases involving overexpression of MMPs, MMP inhibitors (MMPis) have been widely explored [20]. MMP inhibitors have also been utilized as cancer cells imaging agents, due to their selectively and tight binding to MMP receptors. Further, given the presence of fluorine atoms in several derivatives of MMP inhibitors, detection and localization of ^19^F-fluorinated drugs in tumors is possible utilizing magnetic resonance spectroscopy (MRS) [21]. Broad-spectrum inhibition of MMP-1 and other MMP isozymes is believed to be the cause of joint stiffness commonly referred to as musculoskeletal syndrome (MSS) with the chronic dosing of MMPis in both humans and rodents [16].

Considering the promise and advantages of BNCT, there is a need for boron-containing therapeutics that contain a high density of boron atoms and which selectively target tumor cells, as highlighted at a recent National Cancer Institute (NCI) Workshop on Neutron Capture Therapy [22]. Recent reports of new BNCT agents include a receptor-mediated uptake of boron-rich Neuropeptide Y analogues [23]. A membrane-permeable boron cluster was developed using the cell-penetrating lipopeptide pepducin as the vehicle for delivery of boron clusters [24]. Selective tumor cell uptake has also been evaluated for pentagamaboronon-0 (PGB-0) to treat breast cancer [25]. The accumulation of folate in fast-growing cancer cells is exploited by folate receptor-targeting boron compounds in F98 glioma-bearing rats [26]. Nitroimidazole-carborane-modified phenylalanine derivatives as dual-target boron carriers for BNCT targeting of LAT1 and hypoxia [27]. The Beck-Sickinger group has combined tumor-selective small peptides as selective G protein-coupled receptor agonists bearing meta-carborane derivatives enabling targeted delivery to tumor cells [28,29]. Viñas and colleagues have recently reported EGFR inhibitors for BNCT [30] as well as kinase inhibitors [31] which were synthesized using Click chemistry. Herein, we describe new carborane-containing MMP ligands for use in BNCT that bind tightly to MMP enzymes upregulated on the surface of tumor cells without tumor cell penetration.

We have previously described [32] carborane-bearing clusters based on potent and selective α-sulfone hydroxamate MMP inhibitors, SC-78080 [13] and SC-7796419. The first generation of carborane-containing hydroxamate-based MMP inhibitors for use in BNCT were synthesized via thermal Huisgen 1,3-dipolar cycloaddition or copper-mediated azide-alkyne cycloaddition (CuAAC), or the ruthenium-catalyzed azide-alkyne cycloaddition (RuAAC) affording 1,4- and 1,5-triazole compounds containing ortho-closo-carboranes (Figure 1) [32].

Herein, we describe the synthesis of carborane-containing hydroxamate MMP inhibitors based on a broad-spectrum MMP inhibitor CGS-27023A [33] (Figure 1). The IC_50_ values of CGS-27023A inhibition potency at MMP-1, -2, -9, and -13 are 15, 20, 9 and 5 nM, respectively [34]. There are three important elements of the pharmacophore within CGS-27023A: (1) hydroxamic acid, (2) the sulfonamide functional group, and (3) the methoxy aryl hydrophobic tail. These moieties play a key role in pharmacokinetics and binding to the MMP active site [35]. This has led us to explore possibilities of incorporating ortho-*closo* carborane as a direct replacement of isopropyl and pyridine substituent within CGS-27023A molecule without losing substantial potency. These MMP ligands are candidates for BNCT therapy.

## 2. Results

### 2.1. Synthesis

The synthesis of the first BNCT analog begins with tert-butyl D-valine ester hydrochloride **5**, which was used by Ciba-Geigy to synthesize CGS-27023A [33] (Figure 1). In the presence of pyridine, tert-butyl D-valine ester hydrochloride reacted with 4-methoxybenzenesulfonyl chloride to generate **6** in 81% yield. Sulfonamide **6** was reacted with propargyl bromide in DMF in the presence of potassium carbonate to provide alkyne **7** as a white solid in 87% yield. Installation of the ortho-closo-carborane was accomplished through the reaction of alkyne **7** with the activated decaborane complex ^10^B_10_H_12_(MeCN)_2_ prepared from decaborane in toluene with an excess of acetonitrile as the Lewis base under reflux [36]. For this synthesis, enriched decaborane reagent containing >99% of ^10^B isotope was used. The natural abundance of ^10^B in nature is 19.8%, while the natural abundance of ^11^B is 80.2%. Utilization of ^10^B-enriched decaborane will enhance the efficacy of BNCT due to the higher concentration of requisite ^10^B atoms. After reaction of ^10^B_10_H_12_(MeCN)_2_ and the alkyne **7** at 111 °C overnight, carborane 8 was produced. Deprotection of tert-butyl ester 8 was accomplished with TFA in anhydrous methylene chloride followed by azeotroping with toluene to remove excess TFA and volatiles providing carboxylic acid 9 as a white powder. EDC mediated amide coupling reaction of carboxylic acid 9 with O-(tetrahydro-2H-pyran-2-yl) hydroxylamine (THPONH_2_) in the presence of HOBt and N-methylmorpholine (NMM) in DMF afforded THP ether **10**. The THP ether was removed with HCl in 1,4-dioxane and methanol to afford crude product, which purified hydroxamate **3** was isolated in 87% yield via precipitation from the 1,4-dioxane solution.

The synthesis of CGS-23023A analog **4** (Figure 2) was more challenging. Esterification of D-propargylglycine **11** with thionyl chloride in methanol provided the methyl ester **12**. Amine hydrochloride **12** was reacted with 4-methoxybenzenesulfonyl chloride in pyridine, and the yield was significantly enhanced by addition of 4-dimethylaminopyridine (DMAP) which increased the yield providing sulfonamide **13** as a white solid in 75% yield. Alkylation of sulfonamide **13** with 3-picolyl chloride to afford N-picolyl sulfonamide **14** was plagued with a competing base-mediated elimination reaction affording the conjugated species (Z)-methyl 2-penten-4-ynoate and 4-methoxy-N-(pyridin-3-ylmethyl)benzenesulfonamide, based on NMR. Various conditions for the formation of N-picolyl sulfonamide **14** were explored and the best conditions identified were 3-picolyl chloride with cesium carbonate with potassium iodide in DMF to afford alkyne **14** in 34% yield. Saponification of **14** using LiOH provided carboxylic acid **15**. The carboxylic acid 15 was then reacted with O-(tetrahydro-2H-pyran-2-yl) hydroxylamine (THPONH_2_) with EDC, HOBt, and N-methylmorpholine in methylene chloride followed by purification by column chromatography to afford THP-hydroxamate **16** in 65% yield. We initially hoped to react alkyne 16 with the highly activated decaborane complex ^10^B_10_H_12_(MeCN)_2_ to directly form an ortho-closo-carborane, however the presence of the THP-protected hydroxamate was incompatible with the exposure to the Lewis acidic decaborane, consistent with the observations of Kliegel [37], who demonstrated that THP protecting groups can be removed by Lewis acidic boron compounds. Therefore, we employed a Huisgen 1,3-dipolar cycloaddition (Click Chemistry) to install the carborane. The classic high thermal conditions caused elimination of the sulfonamide as observed in the alkylation of sulfonamide **13**. RuAAC conditions employing chloro(pentamethylcyclopentadienyl)(cyclooctadiene)ruthenium(II) gave only unreacted starting material **16** even after extended reaction times at 25 °C, whereas increasing the temperature of the RuAAC reaction to 50 °C gave rise to elimination products. For the CuAAC reaction, copper(II)sulfate pentahydrate was selected due to the literature precedent installing carborane dendrimers, [30,32,38] along with sodium ascorbate in THF:H_2_O (1:1) at room temperature. The reaction progress was monitored through HPLC as well as TLC with PdCl_2_ stain to visualize the carborane. Chromatographic purification afforded the THP-protected 1,4-Click isomer **18** as a white solid. Desilylation to remove the TBDMS group with TBAF and subsequent acidic deprotection of the tetrahydropyranyl hydroxamic acid and addition of the product dissolved CH_2_Cl_2_ to a large volume of diethyl afforded the desired 1,4-triazole **4** as a white solid in 80% yield.

(R)-tert-Butyl 2-(4-methoxyphenylsulfonamido)-3-methylbutanoate (**6**). Sulfonamide **6** was prepared according to the literature, and spectral data match reported values [33].

(R)-tert-Butyl 2-(4-Methoxyphenylsulfonamido-N-(prop-2-yn-1-yl))-3-methylbutanoate (**7**). Alkyne **7** was prepared according to the literature, and spectral data match reported values [39].

Closo-carborane complex from (R)-tert-butyl 2-(4-Methoxyphenylsulfonamido-N-(prop-2-yn-1-yl))-3-methylbutanoate (**8**). To a solution of enriched decaborane, ^10^B_10_H_14_ (3.60 g, 31.5 mmol) in anhydrous acetonitrile (103 mL), anhydrous toluene (271 mL) was added, and the reaction was warmed to reflux for 1 h in a pressurized flask under N_2_. After cooling, alkyne **7** (7.50 g, 19.6 mmol) was added, and the mixture was warmed to 100 °C and stirred for 24 h under N_2_. The reaction’s progress was monitored by HPLC for consumption of starting material. Once the reaction was complete, the mixture was cooled and filtered through a Whatman filter paper type 1 using vacuum filtration. The filter was washed with additional amounts of anhydrous toluene (50 mL), and the filtrate was evaporated under reduced pressure. The crude mixture was then purified by column chromatography (ethyl acetate: hexane = 2:8) yielding tert-butyl ester **8** as a yellow powder (1.66 g, 17%). HPLC analysis: 96.1% AUC. mp 150–155 °C; ^1^H NMR (500 MHz, CDCl_3_) δ 7.78–7.72 (m, 2H), 7.04–6.98 (m, 2H), 4.71 (d, *J* = 17.4 Hz, 1H), 4.66 (s, 1H), 4.13 (d, *J* = 17.5 Hz, 1H), 3.90 (d, *J* = 1.7 Hz, 3H), 3.60 (dd, *J* = 9.5, 1.7 Hz, 1H), 2.50–1.94 (m, 11H), 1.25 (s, 9H), 1.15–1.10 (m, 3H), 0.94 (dd, *J* = 6.7, 1.8 Hz, 3H). 13C NMR (125 MHz, CDCl_3_) δ 169.56, 163.82, 130.64, 114.53, 82.68, 75.65, 66.82, 60.38, 55.75, 51.29, 27.75, and 19.21. HRMS (ESI-ToF): *m/z* calcd for [C1_9_H_37_^10^B_10_NNaO_5_S]^+^ [M+Na]^+^: 514.3578 found 514.3590.

Carboxylic Acid Closo-Carborane **9**. To a solution of tert-butyl ester **8** (1.66 g, 3.38 mmol) in anhydrous methylene chloride (28 mL), trifluoroacetic acid (11 mL) was added, and the mixture was stirred for 2 h at room temperature. The reaction progress was monitored by HPLC for the consumption of starting material. Once the reaction was complete, the solvent was evaporated under reduced pressure, and the traces of trifluoroacetic acid were removed by adding toluene (7 mL) and concentrated at ambient temperature again under reduced pressure to yield carboxylic acid **9** as a white powder (1.56 g, 99%). HPLC analysis: 96.6% AUC. mp 178–180 °C; ^1^H NMR (500 MHz, DMSO-*d*_6_) δ 12.91 (s, 1H), 7.82 (d, *J* = 8.5 Hz, 2H), 7.11 (d, *J* = 8.4 Hz, 2H), 5.04 (s, 1H), 4.54 (d, *J* = 17.4 Hz, 1H), 4.16 (d, *J* = 17.4 Hz, 1H), 3.86 (s, 3H), 3.60 (d, *J* = 9.6 Hz, 1H), 2.70–1.90 (m, 10H), 1.82 (s, 1H), 1.00 (s, 3H), and 0.86 (d, *J* = 6.5 Hz, 3H). ^13^C NMR (125 MHz, DMSO-*d*_6_) δ 171.06, 163.25, 130.46, 128.43, 114.39, 76.33, 66.21, 62.16, 55.69, 50.98, 29.09, 20.97, and 18.96. HRMS (ESI-ToF): *m/z* calcd for [C_15_H_28_^10^B_10_NO_5_S]^−^ [M−H]^−^: 434.2987 found 434.2983.

THP-Protected Hydroxamate Closo-Carborane **10**. To a solution of carboxylic acid **9** (1.40 g, 3.21 mmol) in anhydrous methylene chloride (80 mL), 1-hydroxybenzotriazole hydrate (HOBt, 477 mg, 3.53 mmol), 4-methylmorpholine (NMM, 2.0 mL), O-(Tetrahydro-2-H-pyran-2-yl)hydroxylamine (1.20 g, 9.86 mmol), and *N*-(3-dimethylaminopropyl)-*N*′-ethylcarbodiimide hydrochloride (EDC-HCl, 876 mg, 4.57 mmol) were added. The mixture was vigorously stirred overnight at room temperature under N_2_. The reaction’s progress was monitored by HPLC for the consumption of the starting material. Once the reaction was complete, the reaction mixture was diluted with DI water (80 mL), and extracted with dichloromethane (3×, 80 mL). The combined organic layers were washed with brine and dried over Na_2_SO_4_. The solvent was evaporated under reduced pressure, and the crude was further purified by column chromatography (ethyl acetate: hexane = 1:1) yielding THP-hydroxamate 10 as a white powder (1.26 g, 73%). HPLC analysis: 95.8% AUC. mp 68–70 °C; ^1^H NMR (500 MHz, CDCl_3_) δ 11.06 (s, 1H), 7.89 (d, *J* = 7.7 Hz, 1H), 7.84 (d, *J* = 8.0 Hz, 1H), 7.12–7.06 (m, 2H), 5.24–5.15 (m, 1H), 5.00 (s, 1H), 3.98 (d, *J* = 17.5 Hz, 1H), 3.86 (s, 3H), 3.81 (s, 1H), 3.50 (s, 3H), 2.50–1.90 (m, 10H), 2.14 (s, 1H), 1.48 (s, 5H), 1.24 (s, 1H), 1.10–1.02 (m, 3H), and 0.83 (s, 3H). ^13^C NMR (126 MHz, CDCl_3_) δ 167.05, 164.17, 164.14, 130.21, 114.78, 102.12, 62.18, 55.82, 55.79, 50.90, 29.71, 27.89, 27.78, 24.97, 24.93, and 18.20. HRMS (ESI-ToF): m/z calcd for [C_20_H_37_^10^B_10_N_2_O_6_S]^−^ [M−H]^−^: 533.3672 found 533.3662.

Hydroxamic Acid Closo-Carborane **3**. To a solution of THP-ether **10** (1.15 g, 2.15 mmol), anhydrous dioxane (12 mL), anhydrous methanol (8.0 mL) and 4N HCl in 1,4-dioxane (0.40 mL) were added, and the reaction was stirred at room temperature for 2.5 h. The reaction’s progress was monitored by HPLC for the consumption of the starting material. Once the reaction was complete, the reaction mixture was concentrated under reduced pressure to produce a crude oil. The crude product was dissolved in a minimal amount of dichloromethane (1.0 mL) and methanol (2.0 mL), then slowly pipetted into a stirring solution of hexane (80 mL) and diethyl ether (20 mL). The product precipitated out of the solution and was vacuum filtered through a fine fritted glass filter to obtain hydroxamate 3 as a white powder (845 mg, 87%). HPLC analysis: 95.3% AUC. mp 157–160 °C; ^1^H NMR (500 MHz, DMSO-*d*_6_) δ 10.44 (s, 1H), 8.79 (s, 1H), 7.87 (s, 2H), 7.09 (d, *J* = 8.5 Hz, 2H), 5.34–5.27 (m, 1H), 5.02 (s, 1H), 3.86 (d, *J* = 1.7 Hz, 3H), 3.49–3.44 (m, 1H), 2.20–1.90 (m, 11H), 1.85–1.75 (m, 1H), 1.24 (s, 1H), 1.03 (s, 2H), 0.82–78 (m, 2H), and 0.69 (s, 1H). ^13^C NMR (125 MHz, CDCl_3_) δ 163.76, 130.94, 128.70, 114.90, 79.71, 79.45, 79.19, 70.26, 63.81, 63.27, 62.79, 56.26, 52.92, 49.07, 30.35, 21.86, and 19.24. HRMS (ESI-ToF): m/z calcd for [C_15_H_30_^10^B_10_N_2_NaO_5_S]^+^ [M+Na]^+^: 473.3061 found 473.3106.

Methyl (R)-2-aminopent-4-ynoate hydrochloride (**12**)**.** Amine hydrochloride **12** was prepared according to the literature, and spectral data match reported values [39].

Methyl (R)-2-((4-methoxyphenyl) sulfonamido) pent-4-ynoate (**13**). Sulfonamide **13** was prepared according to the literature, and spectral data match reported values [40].

Methyl (R)-2-((4-methoxy-N-(pyridine-3-ylmethyl) phenyl) sulfonamido) pent-4-ynoate (**14**). To a solution of sulfonamide 13 (300 mg, 1.01 mmol) in anhydrous DMF (6.9 mL), Cs_2_CO_3_ (700 mg, 2.15 mmol) was added, followed by potassium iodide (200 mg, 1.20 mmol) and 3-picolyl chloride hydrochloride (252 mg, 1.54 mmol, recrystallized from ethyl alcohol). The solution was stirred for one day under N_2_ at room temperature. The reaction progress was monitored by HPLC for the generation of the desired product, and the reaction was stopped early to avoid formation of undesired byproducts resulting from the elimination. The mixture was then diluted with DI H_2_O (10 mL) and extracted with EtOAc (3×, 10 mL). The organic layers were combined and washed with 5% NaHCO_3_ (2×, 10 mL), followed by DI H2O (2×, 10 mL), brine (2×, 10 mL), and dried with Na_2_SO_4_. The solvent was evaporated under reduced pressure, and the crude was further purified by column chromatography (ethyl acetate: dichloromethane: hexane = 1:1.16:1.16) to provide *N*-picolylsulfonamide **14** as a colorless oil (135 mg, 34%). HPLC analysis: 97.8% AUC. ^1^H NMR (500 MHz, CDCl_3_) δ 8.55–8.50 (m, 2H), 7.84–7.75 (m, 3H), 7.25 (dd, *J* = 7.9, 4.8 Hz, 1H), 7.00–6.93 (m, 2H), 4.72 (dd, *J* = 8.6, 6.1 Hz, 1H), 4.59 (d, *J* = 16.1 Hz, 1H), 4.48 (d, *J* = 16.2 Hz, 1H), 3.89 (s, 3H), 3.58 (s, 3H), 2.77 (ddd, *J* = 17.2, 6.2, 2.7 Hz, 1H), 2.63 (ddd, *J* = 17.2, 8.6, 2.7 Hz, 1H), and 1.98 (t, *J* = 2.7 Hz, 1H). ^13^C NMR (125MHz, CDCl_3_) δ 169.54, 163.17, 149.66, 149.21, 136.46, 132.17, 131.32, 129.83, 123.30, 114.07, 79.12, 71.80, 58.73, 55.66, 52.50, 47.46, and 21.08. HRMS (ESI): m/z calcd for C_19_H_21_N_2_O_5_S^+^ [M+H]^+^: 389.1166 found 389.1153.

(R)-2-((4-methoxy-N-(pyridin-3-ylmethyl)phenyl)sulfonamido)pent-4-ynoic acid (**15**). To a solution of methyl ester **14** (50 mg, 0.13 mmol) in THF: MeOH: H_2_O in 3:1:1 ratio (650 μL), LiOH (15 mg, 0.63 mmol) was added. The solution was stirred for one day under N_2_ at room temperature. The reaction progress was monitored by HPLC for the consumption of the starting material. Once the reaction was completed, the mixture was then extracted with diethyl ether (2×). This extract was discharged. The aqueous phase was acidified with 2 N hydrochloric acid to pH 2–4, and the resulting mixture was extracted with EtOAc (4×, 2.0 mL). The combined organic layers were dried with Na_2_SO_4_. The solvent was evaporated under reduced pressure to afford carboxylic acid **15** as a white solid (30 mg, 63% yield). HPLC analysis: 98.4% AUC. mp 224–226 °C. ^1^H NMR (500 MHz, DMSO-*d*_6_) δ 8.55 (d, 1H), 8.42 (dd, *J* = 4.8, 1.7 Hz, 1H), 7.81 (dt, *J* = 7.9, 2.0 Hz, 1H), 7.77–7.70 (m, 2H), 7.27 (dd, *J* = 7.9, 4.7 Hz, 1H), 7.06–6.99 (m, 2H), 4.53 (d, *J* = 16.8 Hz, 1H), 4.50–4.42 (m, 2H), 4.04 (q, *J* = 7.2 Hz, 1H), 3.83 (s, 3H), 2.77 (t, *J* = 2.6 Hz, 1H), 2.66 (ddd, *J* = 17.1, 5.4, 2.7 Hz, 1H), 2.39 (ddd, *J* = 17.1, 9.0, 2.6 Hz, 1H), and 2.00 (s, 1H). ^13^C NMR (125 MHz, DMSO-*d*_6_) δ 170.82, 170.46, 162.77, 149.75, 148.49, 136.22, 134.35, 131.97, 130.05, 123.36, 114.44, 81.87, 73.54, 61.20, 60.23, 56.07, 47.15, 21.83, 21.63, 21.24, and 14.56. HRMS (ESI): m/z calcd for C_18_H_19_N_2_O_5_S^+^ [M+H]^+^: 375.1009 found 375.0994.

4-Methoxy-N-((2R)-1-oxo-1-(((tetrahydro-2H-pyran-2-yl)oxy)-l2-azaneyl)pent-4-yn-2-yl)-N-(pyridin-3-ylmethyl)benzenesulfonamide (**16**). To a solution of carboxylic acid **15** (85 mg, 0.23 mmol) in anhydrous methylene chloride (80 mL), 1-hydroxybenzotriazole hydrate (HOBt, 34 mg, 0.25 mmol), 4-methylmorpholine (NMM, 0.4 mL) O-(Tetrahydro-2-H-pyran-2-yl)hydroxylamine (83 mg, 0.71 mmol), and *N*-(3-dimethylaminopropyl)-*N*′-ethylcarbodiimide hydrochloride (EDC-HCl, 63 mg, 0.33 mmol) were added. The mixture was vigorously stirred overnight at room temperature under N_2_. The reaction progress was monitored by HPLC for the consumption of the starting material. Once the reaction was complete, the reaction mixture was diluted with DI water and extracted with dichloromethane (3×). The combined organic layers were washed with 5% NaHCO_3_, DI water, brine, and dried over Na_2_SO_4_. The solvent was evaporated under reduced pressure, and the crude was further purified by column chromatography (ethyl acetate: hexane = 1:1) yielding THP-hydroxamate 16 as a white solid (70 mg, 65%). HPLC analysis: 98.1% AUC. mp 55–58 °C. ^1^H NMR (500 MHz, CDCl_3_) δ 9.42 (s, 1H), 9.33 (s, 1H), 8.59 (dd, *J* = 5.7, 2.3 Hz, 1H), 8.55–8.50 (m, 1H), 7.79 (dt, *J* = 8.0, 2.0 Hz, 1H), 7.78–7.65 (m, 3H), 7.27–7.21 (m, 1H), 6.97–6.90 (m, 3H), 4.98–4.93 (m, 1H), 4.76 (s, 1H), 4.62 (dd, *J* = 19.1, 16.0 Hz, 1H), 4.57–4.47 (m, 2H, 4.17 (q, *J* = 7.1 Hz, 1H), 3.99 (s, 1H), 3.94 (dt, *J* = 10.0, 5.0 Hz, 1H), 3.87 (d, *J* = 0.9 Hz, 3H), 3.75–3.63 (m, 1H), 3.69 (s, 1H), 3.52 (q, *J* = 7.0 Hz, 1H), 2.68 (dddd, *J* = 24.7, 17.4, 6.5, 2.6 Hz, 1H), 2.58–2.46 (m, 1H), 1.86–1.65 (m, 3H), 1.67–1.57 (m, 2H), and 1.31–1.17 (m, 2H). ^13^C NMR (126 MHz, CDCl_3_) δ 171.15, 165.65, 165.51, 163.45, 149.83, 149.80, 149.10, 149.06, 136.55, 136.42, 132.27, 132.19, 130.83, 129.87, 129.83, 123.36, 123.33, 114.27, 102.32, 101.71, 78.80, 71.73, 71.68, 66.38, 65.84, 64.38, 62.20, 62.18, 60.49, 56.90, 56.83, 55,70, 45.89, 45.87, 35.23, 30.65, 29.70, 27.79, 27.75, 24.95, 21.02, 20.76, 19.59, 19.33, 19.13, 18.21, 15.08, 14.20, and 13.71. HRMS (ESI): m/z calcd for C_23_H_28_N_3_O_6_S^+^ [M+H]+: 474.1693 found 474.1685.

TBDMS o-carborane Propyl Azide **17**. TBDMS o-carborane propyl azide **17** was prepared according to the literature, and spectral data match reported values [32].

THP-protected 1,4-Click Isomer **18**. To a solution of THP-hydroxamate **16** (100 mg, 0.21 mmol) in THF: H_2_O (2.0 mL, 1:1 *v*/*v*), sodium ascorbate (6.2 mg, 15 mol %), CuSO_4_·5H_2_O (5.3 mg, 10 mol%) and TBDMS o-carborane propyl azide 17 (72 mg, 0.21 mmol)were added. The mixture was vigorously stirred for 16 h at room temperature under N2. The reaction progress was monitored by HPLC for the consumption of the starting material. Once the reaction was complete, the reaction mixture was quenched by evaporation of the solvent, diluted with DI H_2_O (2.0 mL) and extracted with ethyl acetate (3×, 3.0 mL). The combined organic layers were washed with 10% ammonium chloride (3.0 mL), DI H_2_O (3.0 mL), brine (3.0 mL), and dried over Na_2_SO_4_. The solvent was evaporated under reduced pressure and purified by column chromatography eluting with EtOAc:Hexane (50:50) to afford THP-protected 1,4-triazole isomer **18** as a white solid (167 mg, 97%). HPLC analysis: 96.0% AUC. mp 93–95 °C. ^1^H NMR (500 MHz, CDCl_3_) δ 9.39 (s, 1H), 9.30 (s, 1H), 8.58 (s, 1H), 8.54 (s, 1H), 7.79 (dd, *J* = 13.6, 7.3 Hz, 1H), 7.72 (dd, *J* = 20.1, 8.5 Hz, 2H), 7.26 (t, *J* = 6.2 Hz, 1H), 7.21 (s, 1H), 7.15 (s, 1H), 6.96 (dd, *J* = 8.7, 6.7 Hz, 2H), 4.75 (dd, *J* = 9.5, 4.8 Hz, 1H), 4.74–4.66 (m, 1H), 4.67 (d, *J* = 6.4 Hz, 1H), 4.64–4.55 (m, 1H), 4.30–4.17 (m, 2H), 3.94 (dd, *J* = 16.4, 6.9 Hz, 1H), 3.89 (s, 3H), 3.66 (d, *J* = 10.9 Hz, 1H), 3.64–3.58 (m, 1H), 3.25 (ddd, *J* = 27.0, 14.3, 9.6 Hz, 1H), 2.76 (dt, *J* = 14.4, 5.5 Hz, 1H), 2.62–1.93 (m, 10H), 2.19 (q, *J* = 6.1, 5.0 Hz, 3H), 2.40–2.01 (m, 3H), 1.77 (s, 6H), 1.59 (d, *J* = 11.0 Hz, 2H), 1.31–1.25 (m, 1H), 1.00 (s, 10H), 0.24 (d, *J* = 2.3 Hz, 6H), and 0.09 (s, 1H). ^13^C NMR (125 MHz, CDCl_3_) δ 171.20, 166.26, 166.06, 163.35, 130.68, 129.51, 114.65, 102.00, 101.71, 79.80, 62.14, 62.07, 60.42, 57.61, 57.33, 55.77, 55.75, 49.52, 34.82, 30.70, 30.62, 29.70, 27.76, 27.73, 27.43, 26.44, 26.29, 24.91, 21.07, 20.29, 18.13, 14.20, −2.46, −2.55, and −2.58. HRMS (ESI): m/z calcd for [C_34_H_59_B_10_N_6_O_6_SSi]^+^ [M+H]^+^: 817.4911 found 817.4911.

Hydroxamic acid 1,4-triazole **4**. TBDMS-protected 1,4-triazole **18** (70 mg, 0.086 mmol) was dissolved in anhydrous THF (1.0 mL) and the resulting mixture was cooled to −78 °C. To the cryogenic mixture, a solution of 1M TBAF in THF (100 µL) was added. After 5 min, the cooling bath was removed, and, then, the reaction was allowed to warm to room temperature. After 75 min at room temperature, HPLC analysis showed the complete consumption of the starting material. The reaction mixture was concentrated into a crude oil residue which was dissolved in ethyl acetate (2.0 mL) and washed with DI H_2_O (2.0 mL). The aqueous phase was extracted with ethyl acetate (2×, 2.0 mL). The combined organic layers were washed with DI H_2_O (2.0 mL), dried with Na_2_SO_4_, filtered and concentrated under reduced pressure to produce a crude desired product (61 mg), which was taken forward without any further purification. HPLC purity: 95.7% AUC (retention time: 10.8 min). THP-protected 1,4-triazole from the desilylation step (61 mg) was dissolved in anhydrous 1,4-dioxane (0.5 mL)/methanol (0.1 mL) and allowed to stir under nitrogen atmosphere until complete dissolution was achieved. To the solution, 4 N HCl in 1,4-dioxane (100 µL) was added, and the reaction was allowed to stir for 2.5 h at room temperature where HPLC analysis revealed that the reaction was complete. The reaction mixture was concentrated under reduced pressure to give a crude oil. The crude oil was dissolved in dichloromethane (300 µL), and diethyl ether (2.0 mL) was slowly added to generate a white slurry. The slurry was allowed to stir at ambient temperature for 20 min. The slurry was filtered, and the filter cake was washed with diethyl ether (1.5 mL) followed by hexane (3.0 mL); then, the solids were further dried in vacuo at room temperature to provide the desired 1,4-triazole carboranyl hydroxamic acid hydrochloride salt **4** (45 mg, 80%) as a white solid. HPLC analysis: 96.3% AUC. mp 188–190 °C. ^1^H NMR (500 MHz, DMSO-*d*_6_) δ 11.01 (s, 1H), 8.86 (d, *J* = 8.1 Hz, 1H), 8.06 (s, 1H), 7.80–7.68 (m, 4H), 7.12–7.03 (m, 3H), 5.20 (s, 1H), 4.96 (d, *J* = 17.4 Hz, 1H), 4.82 (d, *J* = 17.5 Hz, 1H), 4.74–4.65 (m, 1H), 4.30 (q, *J* = 7.3, 6.9 Hz, 2H), 3.86 (s, 3H), 3.23–3.14 (m, 2H), 3.07 (dd, *J* = 14.5, 8.4 Hz, 1H), 2.77–2.69 (m, 1H), 2.59 (s, 1H), 2.61–1.77 (m, 10H), 2.33–2.19 (m, 3H), 2.22 (s, 7H), 1.96 (t, *J* = 6.0 Hz, 1H), 1.93 (d, *J* = 7.3 Hz, 1H), 1.58 (ddt, *J* = 16.0, 12.1, 7.1 Hz, 2H), 1.37–1.22 (m, 3H), and 0.94 (t, *J* = 7.4 Hz, 3H). ^13^C NMR (125 MHz, DMSO-*d*_6_) δ 165.57, 163.33, 145.34, 142.02, 140.81, 140.39, 140.01, 130.40, 129.90, 127.02, 123.56, 115.03, 114.86, 76.08, 63.61, 57.14, 56.25, 48.63, 45.24, 33.98, 30.00, 27.08, 23.54, 19.69, and 13.97. HRMS (ESI): m/z calcd for [C_23_H_37_B_10_N_6_O_5_S]^+^ [M+H]^+^: 619.4971 found 619.4971.

### 2.2. MMP Inhibition

The MMP inhibitory potencies of the N-carboranylmethyl analog **3** and 1,4-triazole **4** were assayed and are tabulated in Table 1, alongside the previously reported SC-78080/SC-77964 analogs **1** and **2**. Three target MMP enzymes were selected for the MMP inhibition assay, the collagenase MMP-1, and the gelatinases MMP-2 and MMP-9, which are upregulated in a number of tumor types. NNGH (BML-205) [41] was included as a positive control. The sulfone hydroxamates **1** and **2** are potent to the gelatinases MMP-2 and MMP-9, as reported previously [32], while they lack any inhibition of the collagenase MMP-1. This selectivity was created by design in SC-78080/SC-77964, out of the concern that MMP-1 might be a contributor to the musculoskeletal syndrome (MSS) side effect. The MSS side effect has resulted in clinical trials from chronic dosing of pan MMP inhibitors, but BNCT agents are dosed acutely, so the liability of a compound to evoke MSS upon long-term dosing should be of no concern for short term BNCT usage. The CGS-27023A analogs **3** and **4** were significantly more potent than sulfones **1** and **2** at MMP-2, and notably more potent than the parent CGS-27023A, with reported IC_50_ values for MMP-1, -2, -9, and -13 as 15 nM, 20 nM, 9 nM, and 6 nM, respectively. The CGS analog with the closo-carborane replacing the pyridine moiety has MMP potencies 10× more potent, 14× less potent, and 11× less potent than CGS-27023A at MMP-2, -9, and -13, respectively, which may in part be due to the lower solubility of the nonbasic and hydrophobic carborane, replacing the more polar and ionizable pyridine. The 1,4-triazole CGS-27023A analog **4** fared better, with nanomolar potency for all three isozymes tested, and IC_50_ values for MMP-2, -9, and -13 at 5 nM, 5 nM, and 3 nM, respectively. This is significantly better than CGS-27023A, CGS-27023A analog **3**, and sulfones **1** and **2**.

### 2.3. Molecular Docking

Due to their unique electronic properties and non-classical bonding interactions, including hexacoordinate carbon and boron atoms, carboranes are icosahedral boron-rich clusters that are difficult to model computationally in contrast to traditional organic molecules [42]. Most currently available molecular modeling software packages do not have inbuilt parameters or empirical potential energy functions for various types of boron-atoms and lack the ability to predict drug/ligand interactions for molecules bearing carborane clusters [42,43]. Phenyl rings can act as bioisosteric replacements for carboranes, as can adamantane, due to their similarity in size and lipophilicity, which has been a practical approach to modeling and docking in medicinal chemistry [44]. There are a number of examples of closo-carboranes that have been used as bioisosteric replacements for heteroaromatic or heteroaliphatic rings [45,46,47,48,49,50,51,52,53,54]. As we have reported [32] for other MMP-targeting carborane-containing BNCT candidates, we employed an approach of replacing the carborane moiety with a phenyl ring for molecular docking experiments, given the similar 3-dimensional sweep volume and lipophilicity of both moieties.

The reported solution structure of the known hydroxamate MMP inhibitor **19** (SC-74020) bound to MMP-2 (PDB accession code: 1HOV) [55] was used as a model for the docking experiments. Molecular models employing a phenyl ring in place of the carborane were developed using the Molecular Operating Environment [56] (MOE) computational suite’s Builder utility employing the deprotonated species **20** and **21** specified in Figure 2 as model compounds for carborane MMP ligands **3** and **4**, respectively. Model compound **20** is a known analog of CGS-27023A referred to as CGS-25966.

The docking results of **20** show the compound forming a number of contacts to the catalytic Zn^2+^ ion and surrounding amino acids in a similar manner as for CGS-27023A (Figure 3A,B). The hydroxamate oxygen atom and carbonyl functional group form a bidentate interaction with active site Zn^2+^ (Figure 3). This key interaction is stabilized by additional interactions with side chains of His 120 and His 130. One hydrogen bond is formed between sulfone moiety and Leu 83 on the protein’s backbone (Figure 3A). The longer hydrophobic 4-methoxybenzene tail protrudes deeply into the S1′ subsite, which is in an open conformation. The capacity of the P1′ pocket to accommodate such large moieties as the 4-methoxybenzenesulfonate is rather remarkable and demonstrates that the pocket is both flexible and open at the bottom. The isopropyl group in **20** extends into the solvent, and altering this group should therefore have little impact on the potency of MMP-2. On the other hand, the benzyl moiety occupies a smaller S2′ subsite of the enzyme, which further stabilizes binding through additional hydrophobic interactions with His 120. The binding poses of **20** to MMP-2 were compared with the binding mode of CGS-27023A. The key bidentate interaction with Zn^2+^ in the active site, as well as the hydrogen bond with sulfone moiety, were also observed in the binding of CGS-27023A with MMP-2 (PDB: 1HOV).

The docking results of 1,4-triazole **21** show the compound forming an intricate web of contacts to the catalytic Zn^2+^ ion and surrounding amino acids in a similar manner to CGS-27023A (Figure 4A,B). Comparable bidentate interaction with the active site Zn^2+^ is observed between the hydroxamate oxygen and carbonyl functional group (Figure 4). A key hydrogen bond is formed between the sulfone moiety and Leu 83 on the protein’s backbone (Figure 4A). The longer hydrophobic 4-methoxybenzene tail protrudes deeply into the S1′ subsite, which is in an open conformation. The 1,4-triazole propyl carborane moiety extends into the solvent. On the other hand, the pyridine moiety occupies the smaller S2′ subsite of the enzyme, which further stabilizes binding through a hydrophobic interaction with Tyr 142 (Figure 4B). The binding poses of **21** to MMP-2 were comparable with the binding of CGS-27023A. The key bidentate interaction with Zn^2+^ in the active site, as well as hydrogen bond with sulfone moiety, was observed in the binding of CGS-27023A with MMP-2 (PDB: 1HOV). Overall, dockings of both **20** and **21** illustrate that the installation of carborane on the P2′ and P1 substituents of the CGS-27023A is not expected to affect the pharmacophore properties and to not significantly affect the binding and selectivity of these analogs toward MMP-2. Docking simulations in the MMP-2 active site (PDB: 1HOV) of MMP ligands **1** and **2**, with replacement of the carborane moiety with the isosteric phenyl moiety, have been previously described and reported [32]. The binding properties of phenyl for carborane substituted ligands **1** and **2** show the same key interactions observed with the analogs **3** and **4** and analogs **21** and **22** in the docking studies that include: key H-bond interaction between the oxygen of the sulfone and Leu83 amino acid residue, as well as bidentate interaction between the Zn^2+^ and hydroxamate group on the inhibitor. The piperidine N-substituents of the **1** and **2** surrogates extends into the solvent and allows the incorporation of larger molecular weight functionalities, while the diphenyl ether trail protrudes deeply into the S1′ pocket. Thus, the pharmacophore elements of **1** and **2** are retained in **3** and **4**. Further docking images are included in the Appendix A.

### 2.4. BNCT Results In Vitro BNCT Effect

#### 2.4.1. Cytotoxicity: Evaluation of CGS-27023A Analog **3**

The minimum requirement for our screening protocol of boron compound samples for BNCT is water (and/or DMSO)-soluble 30 mg for first screening of experiments. MMP ligand **3** was eliminated from further evaluation at this step of the protocol due to its toxicity and/or direct (non-BNCT mediated) tumoricidal effect. Acceptable cell toxicity was exhibited for compounds **1** (**B1**) and **2** (**B2**), Figure 5.

#### 2.4.2. Evaluation of SC-78080/SC-77964 Analogs **1** (**B1**) and **2** (**B2**) for BNCT Efficacy

BNCT efficacy is demonstrated in SCC VII cells (Figure 6) and in U87 glioma cells (Figure 7). The survival fraction of **1** (**B1**) and **2** (**B2**) in a pre-incubation boron concentration at 0.143 ppm ^10^B and 0.101 ppm ^10^B, respectively, were similar (Figure 6). The IC_50_ values of **1** and **2** (**B1** and **B2**) were determined to be 2.04 × 10^−2^ mg/mL and 2.67 × 10^−2^ mg/mL, respectively. These values recommend further in vitro/in vivo studies. The IC_50_ of BPA is 2.55 × 10^−2^ M (Figure 5).

In BNCT experiments with glioma U87 delta EGFR cells showed similar survival rates with SCC VII; Survival rates of U87 delta EGFR cells were 24.2 ± 7.7% and 6.2 ± 1.4% for the group treated with 20 ppm of ^10^B in BPA, and 3.0 ± 0.4 % and 0 % for the group treated with 0.7 ppm of ^10^B in hydroxamate **2** (**B2**) when irradiated with thermal neutron fluence of 1.5 × 10^12^ cm^−2^, and 2.7 × 10^12^ cm^−2^, respectively (Figure 7). Treatment with 0.7 ppm of ^10^B in hydroxamate **1** (**B1**) reduced survival rates without neutron irradiation to 23.0% compared with the non-treatment control and 0%, when irradiated with thermal neutron fluence of 1.5 × 10^12^ cm^−2^. The absorbed dose of U87 delta EGFR cells was estimated by comparing the absorbed calculated values of the SCC VII control survival fraction with the control survival fraction of U87 delta EGFR in the same experimental condition. Survival fraction raw data appears in the Appendix A for U87 MG cells and in Appendix A for SCC7 cells.

#### 2.4.3. Evaluation of MMP Carborane **4** for BNCT Efficacy

The survival fraction in U87 glial cells treated with MMP carborane **4** with BPA as a standard is shown in Figure 8, and the survival fraction in SCC VII cells is shown in Figure 9. In U87 MG delta EGFR cells, cell killing effects were similar between MMP carborane **4** and BPA, although the ^10^B concentration in medium was 41.6 times higher in the BPA compared with the MMP carborane **4** (BPA: 20 ppm ^10^B and MMP carborane **4**: 0.48 ppm) (Figure 8). On the other hand, in SCC VII cells, the cell killing effect was much lower in the MMP carborane **4** than in the BPA at the treatment dose in medium (BPA: 20 ppm ^10^B and MMP carborane **4**: 0.48 ppm) (Figure 9). Investigation of the cellular concentration of ^10^B may explain the difference of the two cell lines, therefore, we investigated the cellular concentration of boron by ICP-OES in Figure 10 and Figure 11 as well as of **1** (**B1**) and **2** (**B2**).

### 2.5. Boron Cell Concentration Determination

The boron concentrations of MMP carboranes **1** (**B1**), **2** (**B2**), **4,** and BPA in SCC VII and in U87 delta EGFR cells are shown in Figure 10 and Figure 11, respectively. The boron concentration of BPA (20 ppm ^10^B) and MMP carborane **4** (0.48 ppm ^10^B) in SCCVII cells were 482 ng/E6 cells and 180 (consists of 22% ^10^B, therefore 39.6 ^10^B) ng/E6 cells (Figure 10). Additionally, the **1** (**B1**) and **2** (**B2**) concentrations were very low: 27.8 ng and 19.4 ng, respectively, while the boron concentration of BPA (20 ppm ^10^B) and MMP carborane **4** (0.48 ppm ^10^B) in U87 delta EGFR cells were 452.3 ng/E6 cells and 295.4 (consists of 22% ^10^B, therefore 64.9 ^10^B) ng/E6 cells (Figure 11). As in SCC VII cells, **1** (**B1**) and **2** (**B2**) concentrations were very low in U87 delta EGFR cells. As expected, the difference in concentrations of boron between BPA (20 ppm ^10^B) and MMP carborane **4** (0.48 ppm ^10^B) was smaller in U87 delta EGFR cells than in SCC VII cells. Raw data for boron concentration determination by ICP-OES appears in the Appendix A.

Considering the ^10^B concentration measured by inductively coupled plasma atomic emission spectroscopy (ICP-OES), the relative killing effect is as follows. In the survival fraction of SCC VII cells, the thermal neutron fluence yielding the intrinsic radiosensitivity D37 values (the susceptibility to lethal injury, expressed as the D37 value, which is the radiation dose permitting 37% survival), were 6.0 × 10^11^ for BPA and 2.50 × 10^12^ for MMP carborane **4** (Figure 9). The ^10^B concentration measured by ICP-OES was 482 ng B/E6 cells for BPA and 180.0 × 0.22 ng B/E6 cells for MMP carborane **4**. The relative killing effect of MMP carborane **4** relative to BPA is 2.92, which is calculated by the formula {482/(180 × 0.22)} × (6.0 × 10^11^)/(2.5 × 10^12^). Similarly, in the survival fraction of U87 MG delta EGFR cells, the thermal neutron fluence yielding the D37 values were 1.2 × 10^12^ for BPA and 1.40 × 10^12^ for MMP carborane **4** (Figure 8). The ^10^B concentration measured by ICP-OES was 452.3 ng B/E6 cells for BPA and 295.4 × 0.22 ng B/E6 cells for MMP carborane **4**. The relative killing effect of MMP carborane **4** to BPA is 5.97, which is calculated by the formula {452.3/(295.4 × 0.22)} × (1.2 × 10^12^)/(1.4 × 10^12^). In the case of **2** (**B2**), the thermal neutron fluence yielding D37 values were 1.0 × 10^12^ for BPA and 2.0 × 10^11^ for **2** (**B2**) (Figure 7). The ^10^B concentration measured by ICP-OES was 452.3 ng B/E6 cells for BPA and 31.5 × 0.22 ng B/E6 cells for **2** (**B2**). The relative killing effect of **2** (**B2**) to BPA is 326, which is calculated by the formula {452.3/(31.5 × 0.22)} × (1.0 × 10^12^)/(2.0 × 10^11^).

### 2.6. Water Solubility Determinations 

The water solubilities of test compounds (**1**, **2**, **3**, and **4**) were evaluated using an HPLC-based method, as summarized in Table 2. The two triazoles **1** and **2**, and CGS-27023A analog **3** all have low, single-digit solubility, whereas the CGS-27023A analog triazole **4** exhibits solubility of just under 100 g/mL.

## 3. Discussion

We have synthesized carborane-containing SC-78080/SC-77964 MMP inhibitor analogs **1** and **2** (**B1** and **B2**, respectively) and MMP inhibitor CGS-27023A carborane-containing analogs **3** and **4**. The carborane-containing MMP ligands **1**, **2**, **3**, and **4** exhibit nanomolar potency against gelatinases MMP-2 and MMP-9. The CGS-27023 analog **3** exhibits greater potency based in IC_50_ values at MMP-2 (5.4 nM) versus MMP-9 (125 nM), whereas CGS-27023A 1,4-triazole analog **4** is ~5 nM at both MMP-2 and MMP-9. Both MMP ligands **3** and **4** exhibited good potency against MMP-13 (64 nM and 3 nM, respectively). These MMP ligands are somewhat more potent than sulfone hydroxamate **1** (37 nM and 46 nM at MMP-2 and MMP-9, respectively) and are comparable to sulfone hydroxamate triazole **2** (9.8 nM and 13 nM IC_50_ values for MMP-2 and MMP-9, respectively). Molecular docking to MMP-2 (PDB 1HOV) confirms active site binding similar to SC-78080 and SC-77964 for the sulfone hydroxamates **1** and **2**, and similar to CGS-27023A for analogs **3** and **4**.

All four carborane-containing BNCT compounds exhibit low aqueous solubility of 5–7 µg/mL for hydroxamates **1**-**3**, whereas carborane-containing MMP ligand **4** exhibits aqueous solubility of around 100 µg/mL. The role of aqueous solubility in drug development has been explored in detail [57] and it is noted that exposure of active pharmaceutical ingredients (API) can be ensured by reducing particle size, employing surfactants, and by utilizing solubilization techniques such as solid dispersions. The percentage of compounds in early development (in 2010) with water solubility below 100 μg/mL was 68.4%, while 49.1% of early development compounds were below 10 μg/mL. The goal of new analogs in this series is to enhance water solubility.

Using an infiltration inhibitor as a ligand, we have confirmed a high BNCT effect for analogs **1** and **2** (**B1** and **B2**) in in vitro experiments through our screening tests. The absorbed dose yielding the D37 (dose used to inhibit 63% colony formation) values were 0.27 Gy for 1,4-triazole hydroxamate **1** (**B1**), 0.32 Gy for 1,5-triazole hydroxamate **2** (**B2**), 0.82 Gy for BPA and 1.55 Gy for boron-free control. The relative killing effect of 1,4-triazole hydroxamate 1 (B1) to BPA is 0.82/0.27 = 3.0, and that of 1,5-triazole hydroxamate **2** (**B2**) is 0.82/0.32 = 2.6. The survival fraction of **1** and **2** (**B1** and **B2**, respectively) in a pre-incubation boron concentration at 0.143 ppm, ^10^B and 0.101 ppm ^10^B, respectively, were similar, and these results suggest that **1** and **2** (**B1** and **B2**, respectively) are actively accumulated and/or attached to the SCC VII cells. In future studies, it is necessary to evaluate the infiltration suppression and the BNCT effect by in vivo alpha track experiments. The suppression of infiltration of malignant brain tumor cells is innovative in itself. If the suppression of the infiltration of malignant brain tumor cells can be attained by this strategy, it may also improve the clinical outcome of malignancy that correlates with invasiveness, which warrants indication of this MMP-inhibitor-based modality as a BNCT adjuvant treatment options to be considered.

One significant outcome of this study is that MMP inhibitor scaffolds can serve as BNCT delivery agents through the attachment of boron-rich carborane moieties. The efficacy of MMP carborane **4** may depend on tumor types as shown in our data (Figure 8 and Figure 9), MMP-13 expression may vary among tumor types and should be noted. Furthermore, it is highly significant that MMP carboranes 1 (B1) and 2 (B2) are highly efficacious as BNCT agents despite the fact that the measured boron concentrations determined by ICP-OES in SCC VII and U87 tumor cells treated with these compound are extremely low. This confirms the original hypothesis that the tight binding to enzymes overexpressed on the surface of tumor cells is sufficient for BNCT efficacy, and disproves the long-held assumption that a BNCT agent must enter a tumor cell to elicit efficacy upon irradiation with thermal neutrons.

## 4. Materials and Methods

### 4.1. General Experimental Protocols

All reagents, from Sigma-Aldrich (St. Louis, MO, USA) were used without further purification, unless otherwise noted, and solvents were distilled before use. For column chromatography, RediSep^®^ silver silica gel flash columns (Teledyne Isco, Lincoln, NE, USA) were utilized, and aluminum-backed silica gel 200 μm plates (Sorbent Technologies, Inc., Norcross, GA, USA) were used for thin-layer chromatography (TLC). The 1 H (proton) NMR spectra were obtained at 500 MHz and 13C were obtained at 125 MHz using a 500 MHz Bruker spectrometer (Bruker Corporation, Billerica, MA, USA), with tetramethylsilane (TMS) as the internal standard. NMR spectra were processed using the Mnova version 14.3.2 NMR software program provided by Mestrelab Research (Santiago de Compostela, Spain). Melting points (mp) were taken in open capillaries on a Mel-Temp (Fisher Scientific, Waltham, MA, USA) melting point apparatus and are uncorrected. NNGH was purchased from Enzo Life Science (Farmingdale, NY, USA. The purity of all compounds that were assayed was confirmed to be ≥95% as determined by Agilent 1100 high-performance liquid chromatography (Agilent Technologies, Inc., Santa Clara, CA, USA) employing the mobile phase A = 5% acetonitrile B in water and the mobile phase B = 0.1% TFA in acetonitrile with a gradient of 60% B increasing to 95% over 10 min, holding at 95% B for 5 min, then returning to 60% B and holding for 5 min. High-resolution mass spectral (HRMS) data were obtained at the Integrated Molecular Structure Education and Research Center (IMSERC, Northwestern University, Evanston, IL, USA) on an Agilent 6210A TOF mass spectrometer (Agilent Technologies, Inc., Santa Clara, CA, USA) in the positive ion mode coupled to an Agilent 1200 (Agilent Technologies, Inc., Santa Clara, CA, USA) series high-performance liquid chromatography (HPLC) system. Mobile phase A was 0.05% formic acid in nanopore water and mobile phase B was 0.05% formic acid in acetonitrile, and a gradient of 5% to 90% B in mobile phase A over 5 min was applied. Data were processed using MassHunter software version B.04.00 (Agilent Technologies, Inc., Santa Clara, CA, USA).

### 4.2. Enzyme Inhibition Assays

MMP-2, MMP-9, and MMP-13 were assayed for inhibition using the respective abcam: ab139447, ab139449, and ab139451 Inhibitor Screening Kits. The supplied kits were adaptable to a microplate reader protocol designed for observing a fluorometric change associated with the cleavage of substrate analog at 37 °C. Briefly, each assay or control included a combined volume of 100 µL. Inhibitor concentrations were prepared by two-fold serial dilution in the supplied assay buffer to give in assay concentrations of 0, 0.03, 0.078, 0.156, 0.312, 0.625, 1.25, 2.5, and 5.0 µM for each. The enzymes were then diluted in the assay buffer and 20 µL of each was added to give an assay concentrations of 0.042 nM (MMP-2), 1.9 nM (MMP-9), and 1.86 nM (MMP-13). The Mca-Pro-Leu-Gly-Leu-Dpa-Ala-Arg-NH_2_ substrate was added to a final concentration of 4 µM to initiate the reaction. The reaction was excited at 328 nm and emission detected at 420 nm. Fluorescence was measured every 60 s for a total of 960 s.

### 4.3. Kinetic Analysis of Inhibition Data

For each assay, data were truncated at 1000 s and offsets were applied to bring the initial fluorescence of each assay to zero. Replicate data for assays of individual inhibitor concentrations were plotted in KaleidaGraph (Synergy Software) for comparison and obvious outliers were culled from the analysis. The remaining data were averaged to yield one trace for each inhibitor concentration. The average data were combined into a dataset that included all assayed inhibitor concentrations for each MMP paralog. The datasets were then individually exported as tab-delimited text files for import into KinTek Explorer software and was fitted using numerical integration. The kinetic model used is simplified and assumes catalysis occurs in a single step and that each inhibitor is competitive with respect to the thiopeptide substrate (Mca-Pro-Leu-Gly-Leu-Dpa-Ala-Arg-NH_2_). The model used is shown below where **MMPX** refers to enzyme, **TP** refers to the thiopeptide (indicated above), **PP** represents the product(s) of proteolysis and **I** is the inhibitor. 
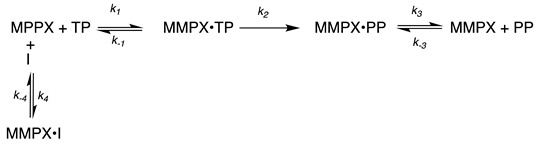


The fitting attempted to define a ratio of k_−4_/k_4_ that defines the dissociation constant for each inhibitor with each MMP host. Convergence of the fit to provide reasonable estimates for the k_−4_/k_4_ ratio was aided by defining the five other rates in the model from published data for each of the enzymes tested. For MPP2 k_2_ was defined from the turnover number (kcat) of 3.8 s^−1^ defined by Olson et al., [58] and k^−1^/k_1_ was defined from the published Km value of 3.0 µM from the same study using the arbitrary rate constant ratio of 60 s^−1^/20 µM^−1^s^−1^. The dissociation constant for product(s) was assumed to be relatively high and the ratio of k_−3_/k_3_ was defined by arbitrary rate constants of 15,000 s^−1^/2 µM^−1^s^−1^ to give a high mM binding constant. Similarly, for MMP9, k_2_ was defined from the turnover number of 4.4 s^−1^ and k_−1_/k_1_ was defined from the published K_m_ value of 2.5 µM from the same study [58], using the arbitrary rate constant ratio of 49 s^−1^/20 µM^−1^s^−1^. The dissociation constant for product(s) was again assumed to be high and the ratio of k_−3_/k_3_ was again defined as 15,000 s^−1^/2 µM^−1^s^−1^. For MMP13, k_2_ was defined from the turnover number of 4.3 s^−1^ from the work of Meinjohanns et al. on mouse MPP13, and k_−1_/k_1_ defined from the published K_m_ value of 5.8 µM from the same study [59], using the constant rate ratio of 87 s^−1^/20 µM^−1^s^−1^. The dissociation constant for product(s) was again assumed to be high and the ratio of k_−3_/k_3_ was defined as it was for MMP-2 and MMP-9. The basis for use of the published mouse kinetic parameters was that the mouse and human enzymes have similar published k_cat_/K_m_ values [60].

In each case, the data converged to give a well-defined ratio for k_−4_/k_4_ that approximates the binding constant for each inhibitor. The error was calculated by propagating the error for ratios of rate constants derived from fitting.

### 4.4. Molecular Docking Protocol

The Molecular model of **20** and **21** was developed using the Molecular Operating Environment (MOE) computational suite’s Builder utility followed by minimization in the gas phase using the MMFF94X force field. The solution structure of compound **19** bound to MMP-2 was then uploaded into MOE and prepared for docking using MOE’s Structure Preparation utility. The hydrogen-bonding network of the docking model was further optimized at pH of 7.4 by automatically sampling different tautomer/protomer states using Protonate3D, which calculates optimal protonation states, including titration, rotamer, and “flips” using a large-scale combinatorial search. The binding pocket of MMP-2 was surveyed using MOE’s Site Finder utility, which employs an alpha shape construction algorithm to identify regions of tight atomic packing in which are filled inactivated dummy atoms that ligand docking can be directed to. Following the preparation of the MMP-2 docking model and inactivation of other atoms, molecular docking using the previously generated ligand conformation database was carried out at the docking site specified by the dummy atoms populating the binding pocket of MMP-2 of the active docking site. Ligand placement employed the Alpha Triangle method with Affinity dG scoring to generate 100 data points per unique ligand that were further refined using the Induced Fit method with GBVI/WSA dG scoring to obtain the top 50 docking poses per ligand. The Amber12:EHT force field was used to perform these calculations.

### 4.5. Cytotoxicity: IC_50_

The IC_50_ (moles/liter), i.e., the concentration which inhibits the growth of SCC VII tumor cells by 50% after 3 days of continuous exposure with the compounds in the concentration gradient manners, was determined. Suspensions of 105 SCC VII cells/200 μL MEM containing 10% FCS (fetal calf serum) were incubated for 3 days in 96-well microplate with various concentrations of **1** and **2** (**B1** and **B2**). After 3 days of exposure, the cell were washed with PBS and fixed with 70% ethanol. The SCC VII cells were stained with 5% Giemsa solution for 15 min. After removing the solution, the cells were dried in air and Giemsa stained cells were uniformized by the addition of 100 μL of alcohol. The optical absorbance at 595 nm was measured by optical densitometer. The IC_50_ was determined as the dose producing a 50% density reduction in the optical absorbance.

### 4.6. Survival Study: Colony Formation Assay

SCC VII tumor cells were incubated and stabilized in a logarithmic subconfluent growth phase at 36 °C in a 5% carbon dioxide atmosphere in a 10 cm diameter tissue culture dishes (100 mm tissue culture dishes; Corning Glass Works, Corning, NY, USA) containing Eagle’s MEM containing 10% heat-inactivated fetal bovine serum and 2 mM L-glutamine. The U87 delta EGFR glioma cell line, kindly provided by Dr. H. Michiue (Okayama University, Okayama, Japan), which stably expressed the constitutively active EGFRvIII, was used in the colony formation assay. The cells were maintained in Dulbecco’s modified Eagle’s medium (DMEM; FUJIFILM Wako Pure Chemical Corporation, Richmond, VA, USA with 10% fetal bovine serum (FBS), penicillin, and streptomycin at 37 °C in a humidified atmosphere containing 5% CO2. The stock solutions of **B1** and **B2** were obtained by dissolving 6 mg of them in 1 ml DMSO, respectively. For SCC VII cells, after adding 100 μL of a boron solution, the cells were incubated for boron loading into tumor cells for 5 h. For U87 delta EGFR cells, they were incubated with 0.7 ppm of final ^10^B concentration in medium. The ^10^B (0.143 ppm ^10^B for **1** (**B1**) and 0.101 ppm for **2** (**B2**) in the medium was reconfirmed by PGS (prompt γ spectroscopy) in the SCC VII colony forming assay. After boron loading, the cells were trypsinized and washed three times by PBS, and 5 × 10^3^ cells/mL MEM (FCS+) were irradiated with thermal neutrons in column-shaped Teflon tubes (1 × 3 cm). The cells did not adhere to the tubes, and no secondary radiation was caused by bombardment with the thermal neutrons. The irradiation times were 0, 15, 30 and 45 min. The thermal neutrons flux was 1.8 × 10^9^ cm^−2^·s^−1^. The thermal neutron fluence was determined by averaging two gold foils symmetrically attached to the surface of the Teflon tube along the direction of the incidence of the thermal neutrons. The neutron absorbed dose (Gy) was calculated using the flux-to-dose conversion factor [61]. The chemical composition of the tumors was assumed to be 10.7% hydrogen, 12.1% carbon, 2% nitrogen, 71.4% oxygen, and 3.8% others [62]. The γ-ray dose rate, including secondary γ-rays, was 1.0 × 10^−2^ Gy·min^−1^, according to a thermoluminescence dosimeter attached to the surface of a Teflon tube containing 1 mL MEM. After thermal neutron exposure, 300 or 900 cells were placed in three Corning 60 mm tissue culture dishes containing 6 mL MEM to examine colony formation. Ten days later, the colonies were fixed with 70% ethanol and stained with 5% Giemsa solution for quantitative visualization by the naked eye.

### 4.7. Statistical Analysis

Statistical analysis and drawings of graphs were carried out via Prism 9.4.1 for macOS, GraphPad Software, LLC (Boston, MA, USA) and with Microsoft Excel for Office 365 MSO, 64-bit, Version 1902 (Microsoft Corporation, Redmond, WA, USA). The in vitro BNCT was carried out in triplicate to reduce the variation. Values are presented as the mean ± SEM from three independent experiments. The significance of the differences in survival rates was assessed by Student’s *t*-test.

### 4.8. Water Solubility

The drug water solubility was determined using Agilent 1260 Infinity II preparative high-performance liquid chromatography (HPLC) and analyzed with ChemStation software (Agilent Technologies, Santa Clara, CA, USA). A set of serial dilutions, in either acetonitrile or methanol, of each of the BNCT analogs was prepared with the following concentrations: 2.0 mg/mL, 1.0 mg/mL, 0.50 mg/mL, 0.25 mg/mL, 0.10 mg/mL, 0.05 mg/mL, and 0.01 mg/mL. The aqueous, unknown concentration, samples of each of the BNCT analogs were prepared by “dissolving” 2.0 mg quantity of the BNCT analogs in 1.0 mL of nanopore, HPLC water. The whole sample did not dissolve in the solution, and the sample was filtered to leave the undissolved solid behind and obtain a clear aqueous solution of unknown concentration. For the HPLC analysis, each serial dilution concentration was prepared and ran in duplicates following three blank runs with either acetonitrile or methanol depending on the sample. In addition, the aqueous solutions of unknown concentration of BNCT agents were also prepared and ran in duplicates following three blank runs with nanopore water. The data from the area under the curve (AUC) were collected at 220 nm and a calibration curve was generated using Microsoft Excel by plotting the concentration of serial dilutions on the x-axis and AUC on the y-axis. From the calibration graph, the linear regression line equation was obtained and the aqueous unknown concentration of the BNCT agent was determined by solving the concentration using the y = mx + b equation based on the generated AUC data at 220 nm of the aqueous sample. See the Appendix A for the raw data and calibration curves (Appendix A).

### 4.9. Intracellular Uptake of Boron Compounds into SCC VII Cells and U87MG Delta EGFR Cells Evaluated with ICP-OES

SCC VII cells and U87MG delta EGFR cells were seeded on 10 cm dishes in cell culture mediums and incubated in 37 °C under 5% CO_2_ incubator until 70~80% confluent. They were treated with boron compounds **1** (**B1**), **2** (**B2**), **4,** or BPA in cell culture mediums for 4 h. The concentration of 10B in the medium was 0.35 or 0.7 ppm for **B1** and **B2**, 0.48 ppm or 2.42 ppm for compound **4** and 20 ppm for BPA. A stock solution of ^10^B-para-boronophenylalanine (BPA) was used for all experiments. BPA was purchased from KatChem Ltd. (Prague, Czech Republic) and prepared by dissolving it in distributed water as a complex with 3% fructose for BPA. The ^10^B concentrations were measured by means of prompt gamma-ray spectrometry using a thermal neutron guide tube installed at the Kyoto University Research Reactor (KUR), and the value was about 1000 ± 4.55 ppm. To count the number of cells after treatment with the boron compounds, the cells (*n* = 1) were washed with PBS, detached by trypsin, and counted with a hemocytometer. For the measurement of boron uptake, the cells (*n* = 3) were washed with PBS (3 mL × 3) and digested with 60% HNO_3_ aq (1 mL) at room temperature for 24 h, which were transferred to 15 mL centrifuge tubes with Milli-Q water (4 mL). These tubes were centrifuged at 3000 rpm and 4 °C for 10 min, and the resulting sample solutions were filtered. The concentration of boron atoms was determined by ICP-OES (ICPE-9000, Shimadzu, Kyoto, Japan). All reagents were purchased from FUJIFILM Wako Pure Chemical Corporation (Osaka, Japan).

## Data Availability

Not applicable.

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
