# Peer review of "Carborane-Containing Hydroxamate MMP Ligands for the Treatment of Tumors Using Boron Neutron Capture Therapy (BNCT): Efficacy without Tumor Cell Entry"

_ijms, 2023, doi:10.3390/ijms24086973_

Round 1
Reviewer 1 Report
The manuscript describes the development of novel carborane-containing MMP ligands as BNCT agents. The authors discovered that carborane derivatives 3 and 4 showed a significant MMP-2 inhibitory potency similar to NNGH. Further, significant in vitro BNCT effects have been observed against SCCVII and U87 glioma cells, that are target tumors of BNCT. This reviewer basically recommends the manuscript suitable for this journal. However, the following issues need to be addressed before acceptance:
1) Docking simulations should be performed on carborane derivatives 3 and 4 and compared to compounds 1 and 2.
2) Boron accumulations of compounds 3, 4, and BPA in SCCVII and for U87 cells should be determined, and the relationship between boron concentrations and BNCT efficacy should be discussed.
3) Due to the requirement of water-solubility of BNCT agents, the authors must determine the water solubility of compounds and discuss the maximum possible dose based on their water solubility.
Author Response
Please see attached reply.

Reviewer 2 Report
The manuscript presented by Becker reports the syntheses of 4 MMP ligand-bearing derivatives on a Carborane cluster, which provides them with the key to a BNCT treatment. . The results are encouraging for future development as agents for BNCT. The work is very well written and comprehensive and needs to be published with minor revisions:
-in the title I would suggest avoiding the use of "cancers", it is too generic,
-In the abstract line 19 specify the abbreviations (i.e. BPA)
_the keywords should not report the same words present in the test.
-line 133, excess TFA...add " and volatiles".
-line 138 from precipitation (which solvent?)
-line 393 CDCl3
-check the molecular formulas in the experimental procedures (numbers have to be written subscripted)
- In the characterization of new compounds, when a stereocenter is present, polarimeter analysis has to be done.
Author Response
Please see the attached reply.

Round 2
Reviewer 1 Report
I read the revised manuscript and found that the authors have addressed most of my comments. I strongly recommend that the survival curve in Figures 8 and 9 should be presented in a similar format to Figure 7 before publication.
Author Response
We have reformatted the Figures so that now Figures 6, 7, 8, and 9 are all formatted in the identical manner. We appreciate the suggestion, and we were aware of the awkward difference in styles of presentation, which came from different authors in their own styles. We have now standardized these Figures.